# SiMHand: Mining Similar Hands for Large-Scale 3D Hand Pose Pre-training

**Nie Lin**[1,*], **Takehiko Ohkawa**[1,*], **Yifei Huang**[1,†], **Mingfang Zhang**[1], **Minjie Cai**[2,†], **Ming Li**[1],
**Ryosuke Furuta**[1] **& Yoichi Sato**[1]
[1]The University of Tokyo, [2]Hunan University
`{nielin,ohkawa-t,hyf,mfzhang,mingli,furuta,ysato}@iis.u-tokyo.ac.jp`,
`caiminjie@hnu.edu.cn, li-ming948@g.ecc.u-tokyo.ac.jp`

## Abstract

We present a framework for pre-training of 3D hand pose estimation from in-the-wild hand images sharing with similar hand characteristics, dubbed **SiMHand**. Pre-training with large-scale images achieves promising results in various tasks, but prior methods for 3D hand pose pre-training have not fully utilized the potential of diverse hand images accessible from in-the-wild videos. To facilitate scalable pre-training, we first prepare an extensive pool of hand images from in-the-wild videos and design our pre-training method with contrastive learning. Specifically, we collect over 2.0M hand images from recent human-centric videos, such as *100DOH* and *Ego4D*. To extract discriminative information from these images, we focus on the *similarity* of hands: pairs of non-identical samples with similar hand poses. We then propose a novel contrastive learning method that embeds similar hand pairs closer in the feature space. Our method not only learns from similar samples but also adaptively weights the contrastive learning loss based on inter-sample distance, leading to additional performance gains. Our experiments demonstrate that our method outperforms conventional contrastive learning approaches that produce positive pairs solely from a single image with data augmentation. We achieve significant improvements over the state-of-the-art method (PeCLR) in various datasets, with gains of 15% on Frei-Hand, 10% on DexYCB, and 4% on AssemblyHands. Our code is available at `https://github.com/ut-vision/SiMHand`.

## 1 Introduction

Hands serve as a trigger for us to interact with the world, as seen in various human-centric videos. The precise tracking of hand states, such as 3D keypoints, is crucial for video understanding (Sener et al., 2022; Wen et al., 2023), AR/VR interfaces (Han et al., 2022; Wu et al., 2020), and robot learning (Chao et al., 2021; Qin et al., 2022). To this end, 3D hand pose estimation has been studied through constructing labeled datasets (Ohkawa et al., 2023a; Zimmermann et al., 2019; Chao et al., 2021; Ohkawa et al., 2023b) and advancing supervised pose estimators (Cai et al., 2018; Ge et al., 2019; Park et al., 2022; Liu et al., 2024; Fan et al., 2024). However, utilizing large-scale, unannotated hand videos for pre-training remains underexplored, while collections of human-centric videos, like 3,670 hours of videos from Ego4D (Grauman et al., 2022) and 131-day videos from 100DOH (Shan et al., 2020), are readily available.

In pre-training, contrastive learning has been utilized to learn from unlabeled images like Sim-CLR (Chen et al., 2020), which maximizes agreement between positive pairs while repelling negatives. Spurr *et al.* (Spurr et al., 2021) introduce pose equivariant contrastive learning (PeCLR) for 3D hand pose estimation, which aligns the geometry of features encoded from augmented images with affine transformations. However, both SimCLR and PeCLR create positive pairs from a single sample by applying data augmentation, limiting the gains from positive pairs as their hand appearance and backgrounds are identical. Ziani *et al.* (Ziani et al., 2022) extend the contrastive learning framework to video sequences by treating temporally adjacent hand crops as positive pairs.

---

* Equal contribution. † Corresponding author.

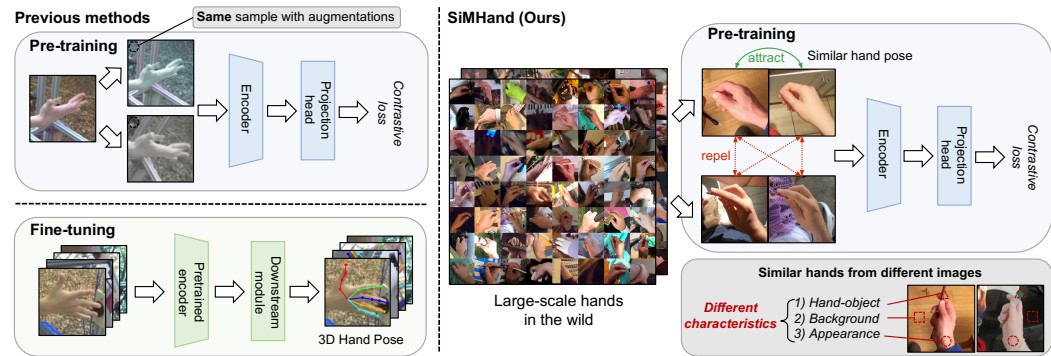

Figure 1: **The pipeline of pre-training and fine-tuning. (Left)** Previous pre-training methods (*e.g.*, PeCLR (Spurr et al., 2021)) learn from positive pairs originating from the different augmentations and fine-tune the network on a dataset. **(Right)** Our method is designed to learn from positive pairs with similar foreground hands, sampled from a pool of hand images in the wild.

However, in-the-wild videos can challenge tracking hands across frames, especially in egocentric views where hands are often unobservable due to camera motion. Meanwhile, this temporal positive sample mining remains the limited appearance variation of hands and backgrounds.

In this work, we introduce SiMHand, a novel contrastive learning framework for 3D hand pose pre-training, which leverages diverse hand images in the wild, with the largest 3D hand pose pre-training set to date. We specifically collect 2.0M hand images from human-centric videos, from Ego4D (Grauman et al., 2022) and 100DOH (Shan et al., 2020), using an off-the-shelf hand detector (Shan et al., 2020). Our pre-training set significantly exceeds the scale of prior works by two orders of magnitude, such as over 32-47K images in (Spurr et al., 2021) and 86K images from 100DOH in (Ziani et al., 2022).

Our method focuses on learning discriminative information by mining hands with similar characteristics from various video domains. Based on our observations, contrastive learning can further benefit from discriminating the foreground of hands in varying backgrounds. As shown in Fig. 1, our positive pairs are sourced from different images, offering additional information gains from different types of object interactions, backgrounds, and hand appearances. Specifically, we use an off-the-shelf 2D hand pose estimator (Lugaresi et al., 2019) to identify similar hands from the pre-training set.

Using the identified similar hands as positive pairs, we further propose adaptive weighting, to dynamically find informative pairs during training. A naive adaptation of the similar hands is to replace the original positive pairs in contrastive learning, but this scheme struggles to exploit *how similar the paired hands are*. To tackle this, we assign weights based on the similarity scores within the mini-batch in the contrastive learning loss. The weights are designed to have higher values as the similarity of the pairs increases. This allows the optimization of contrastive learning to explicitly consider the proximity of samples, beyond binary discrimination between positives and negatives.

We validate the effectiveness of the pre-trained networks by fine-tuning on several datasets for 3D hand pose estimation, namely FreiHand (Zimmermann et al., 2019), DexYCB (Chao et al., 2021), and AssemblyHands (Ohkawa et al., 2023b). Our proposed method consistently outperforms conventional contrastive learning methods, SimCLR and PeCLR. Additionally, we conduct extensive ablation experiments to analyze: 1) performance with varying pre-training and fine-tuning data sizes, 2) the effect of adaptive weighting, and 3) the improvement with different levels of similarity.

In summary, the main contribution of this paper is threefold:

- We propose SiMHand, a contrastive learning method for 3D hand pose pre-training, leveraging positive samples with similar hands mined from 2.0M in-the-wild hand images.

- We introduce a parameter-free adaptive weighting mechanism in the contrastive learning loss, enabling optimization guidance according to the calculated similarity.

- Our experiments demonstrate that our approach surpasses prior pre-training methods and achieves robust performances across different hand pose datasets.

## 2 RELATED WORK

**3D hand pose estimation:** The task of 3D hand pose estimation aims to regress 3D hand joints. Since annotating 3D hand poses is challenging, only limited labeled datasets are available (Ohkawa et al., 2023a), and most of which are constructed in controlled laboratory settings (Zimmermann et al., 2019; Chao et al., 2021; Moon et al., 2020; Ohkawa et al., 2023b). Given this challenge, two approaches have been proposed to facilitate learning from limited annotations: pseudo-labeling and self-supervised pre-training. Pseudo-labeling methods learn from pseudo-ground-truth assigned on unlabeled images (Chen et al., 2021c; Zheng et al., 2023; Liu et al., 2021; Yang et al., 2021; Ohkawa et al., 2022; Liu et al., 2024). For example, S2Hand (Chen et al., 2021c) attempts to learn 3D pose only from noisy 2D keypoints on a single-view image, while HaMuCo (Zheng et al., 2023) extends such self-supervised learning to multi-view setups. Alternatively, pre-training methods aim to find well-initialized models with unlabeled data for downstream tasks. Prior works propose contrastive learning approaches but rely on relatively small pre-training sets (*e.g.*, 32-47K images in (Spurr et al., 2021) and 86K images in (Ziani et al., 2022)). We collect hand images from large human-centric datasets such as Ego4D (Grauman et al., 2022) and 100DOH (Shan et al., 2020), expanding our pre-training set to 2.0M images.

**Contrastive learning:** Contrastive learning has emerged as a powerful technique in self-supervised learning, bringing positive samples closer while pushing negative samples apart (Chopra et al., 2005; Schroff et al., 2015; Song et al., 2016; Sohn, 2016; He et al., 2020; Huang et al., 2023). Standard methods generate positive samples from an identical image with data augmentation (*i.e.*, self-positives) (Grill et al., 2020; Caron et al., 2020; Chen & He, 2021; Radford et al., 2021; Caron et al., 2021), thus the positive supervision doesn't explicitly model inter-sample relationships. To address this, Zhang *et al*. propose a relaxed extension of self-positives, *non-self-positives* (Zhang et al., 2022), which share similar characteristics but originate different images, such as images capturing the same scene (Arandjelovic et al., 2016; Ge et al., 2020; Berton et al., 2022; Hausler et al., 2021), the same person ID (Chen et al., 2021a;b), and multi-view images (Jie et al., 2024). The positive supervision from non-self-positives enables considering diverse inter-sample alignment and facilitates the learning of semantics more easily. Zhang *et al*. identify non-self-positives by searching similar human skeletons from single-view images and adapt in action recognition (Zhang et al., 2022). Jie *et al*. rely on multi-view (*i.e.* paired) images to define non-self-positives and propose pair-wise weights to adaptively leverage useful multi-view pairs (Jie et al., 2024). Our work proposes the mining of non-self-positives from 2D keypoint cues with additional pair-wise weighting to account for similarity from *unpaired* data in pre-training.

## 3 METHOD

Our approach SiMHand aims to pre-train an encoder for 3D hand pose estimation with large-scale human-centric videos in the wild. We first construct a pre-training set from egocentric and exocentric hand videos (Sec. 3.1). Then, we find similar hand images to define positive pairs across videos (Sec. 3.2). Finally, we incorporate these positive pairs into a contrastive learning framework and employ adaptive weights to improve the effectiveness in pre-training (Sec. 3.3).

### 3.1 DATA PREPROCESSING

Our preprocessing involves creating a set of valid hand images for pre-training, which is sampled from a set of $N$ videos: $\{v_1, v_2, \ldots, v_N\}$. We use an off-the-shelf hand detector (Shan et al., 2020) to select valid frames with visible hands. Given a video frame $I_{\text{full}} \in v_i$, the model detects the existence of the hand and its bounding box, creating hand crops enclosing either hand identity (right/left) from $I_{\text{full}}$. To avoid bias related to hand identity, we balance the number of right and left hand crops equally and then convert all crops to right-hand images. Then, we create a set of frames for each video $v_i$ as $\mathcal{F}_i = \{I_{i,1}, I_{i,2}, \ldots, I_{i,T_i}\}$, where $I_{i,j} \in \mathbb{R}^{H \times W \times 3}$ represents the processed crop with height $H$ and width $W$, and $T_i$ is the total number of crops in $v_i$. The height $H$ and width $W$ are defined post-resize to give the uniform image size. Using this frame set $\mathcal{F}_i$, the video dataset can be re-represented as $\mathcal{V} = \{\mathcal{F}_1, \mathcal{F}_2, \ldots, \mathcal{F}_N\}$. Specifically, we processed two datasets, Ego4D (Grauman et al., 2022) and 100DOH (Shan et al., 2020), to collect 1.0M images from 8K and 21K videos, respectively. More details about our preprocessing can be found in the supplement.

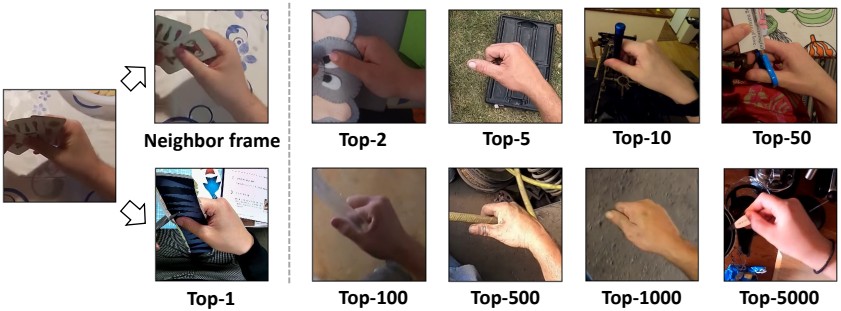

Figure 2: **Visualization of similar hand samples in Top-K.** Given the query image ($I$), the mined similar samples are shown ("Top-1" corresponds to $I^+$ in Sec. 3.2).

## 3.2 MINING SIMILAR HANDS

To incorporate diverse samples in contrastive learning, we design positive pairs from non-identical images with similar foreground hands. Here we construct a mining algorithm to find similar hands from $\mathcal{V}$ by focusing on pose similarity between hand images. We first extract 2D keypoints from $I$, embed in the feature space, and search a positive sample.

**Pose embedding:** We adopt estimated 2D keypoints (for 21 joints) to find similar hands. We use an off-the-shelf 2D hand pose estimator $\phi$ (Lugaresi et al., 2019), but the outputs are prone to be noisy in testing in the wild. To make it more robust, we obtain a $D$-dimensional embedding of 2D hand keypoints, $\mathbf{p} \in \mathbb{R}^D$, for each image $I$. This serves to reduce the noise effect while preserving the semantics of hands. We use a concatenated 42-dimensional vector as the output of $\phi$ for later use. Particularly, we apply PCA-based dimension reduction, which projects the keypoints vector into a lower-dimensional space of size $D$. Given the PCA projection matrix $M \in \mathbb{R}^{42 \times D}$, the pose embedding $\mathbf{p}$ is calculated as $\mathbf{p} = M^T \phi(I)$.

**Mining:** This step is designed to identify a positive sample $I^+ \in \mathbb{R}^{H \times W \times 3}$ paired with a query image $I$. We denote the similarity mining logic as $I^+ = \text{SiM}(I)$. As shown in Fig. 2, using the closest (neighbor) sample in the PCA space encounters a trivial solution $I, I^+ \in v_i$, where both images originate from the same video $v_i$. Similarly to (Ziani et al., 2022), the supervision by neighbor samples of the same video has less diversity in backgrounds, hand appearances, and object interactions. Thus we are motivated to find similar hands derived from different videos. Specifically, we search the minimum distance within the set of all frames except for $v_i$, written as $\mathcal{F}_i^c = \bigcup_{k \neq i} \mathcal{F}_k$. Given an query $I_{i,j}$, which represents the $j$-th image of the $i$-th video, the function $\text{SiM}(\cdot)$ is formulated as

$$\text{SiM}(I_{i,j}) = \arg\min_{x \in \mathcal{F}_i^c} D(M^T \phi(x), M^T \phi(I_{i,j})),\tag{1}$$

where $D(\cdot, \cdot)$ is the Euclidean distance metric.

As a proof of concept, we illustrate examples after our mining $\text{SiM}(\cdot)$ in Fig. 2. We denote "Top-1" (most similar) as our assigned positive sample $I^+$ to the query image $I$. As references, the rest of the figures ("Top-K") represent the $K$-th similar samples. Our sampling highlights the diversity in captured environments and interactions, while it also suggests that as the rank (distance) increases, the sampled images become dissimilar. Additional visualization results of similar hands can be found in supplement.

## 3.3 CONTRASTIVE LEARNING FROM SIMILAR HANDS WITH ADAPTIVE WEIGHTING

We detail our contrastive learning approach (see Fig. 3), learning from mined similar hands with adaptive weighting.

**Overview:** The contrastive learning is designed to align positive samples $(I, I^+)$ in the feature space, constructed in Sec. 3.2, and the rest of negative samples are pushed apart. Following (Chen et al., 2020; Spurr et al., 2021), we treat all mini-batch samples other than the corresponding positive samples as negative samples $I^-$. Feature extraction is performed by two learnable compo-

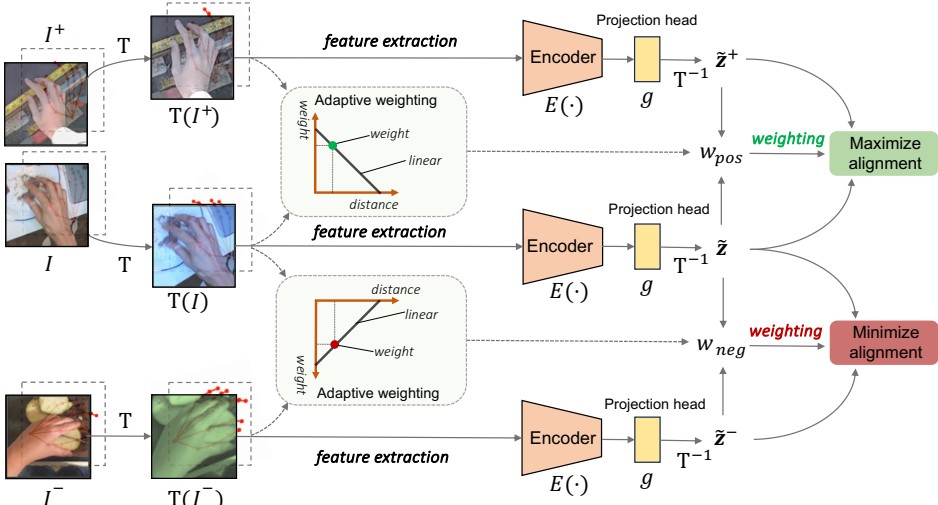

Figure 3: **Overview of our SiMHand.** Starting from the left, hand images ($I$, $I^+$, $I^-$) and their corresponding 2D keypoints are input to the model. After applying random augmentations through transformation $\mathbf{T}$, both the images and 2D keypoints are spatially transformed. The altered 2D keypoints are then used to compute adaptive weights $w_{pos}$ and $w_{neg}$, which guide contrastive learning by strengthening or weakening the alignment between positive and negative samples.

nents: an encoder $E(\cdot)$ and a projection head $g(\cdot)$, which indicates the entire model as $f = g \circ E$. The extraction is combined with image augmentation $\mathbf{T}$, which formulated as $\mathbf{z} = f(\mathbf{T}(I))$ and $\mathbf{z}^+ = f(\mathbf{T}(I^+))$. Applying geometric transformations (*e.g.*, rotation) in $\mathbf{T}$ can cause misalignment between the image and feature spaces; we correct such an error with the inverse transformation $\mathbf{T}^{-1}$ as (Spurr et al., 2021). After applying the inverse transformation to the feature $\mathbf{z}$, we obtain a feature $\tilde{\mathbf{z}} = \mathbf{T}^{-1}(\mathbf{z})$, where geometry is aligned to the original images. As such, all anchor, positive, and negative samples are encoded as $\tilde{\mathbf{z}}$, $\tilde{\mathbf{z}}^+$, and $\tilde{\mathbf{z}}^-$, respectively.

**Adaptive weighting:** During learning from our similar hands, we propose an adaptive weighting per pair to focus more on informative samples that provide greater discriminative information. The assigned weights are computed by the predefined similarity metric in Sec. 3.2. Given pre-processed keypoints for two samples within the mini-batch, $\mathbf{k}_1$, $\mathbf{k}_2$, the weight $w$ is computed by linear scaling with the Euclidean metric $D(\cdot, \cdot)$ as

$$w = \frac{d_{\max} - D(\mathbf{k}_1, \mathbf{k}_2)}{d_{\max} - d_{\min}}, \tag{2}$$

where $d_{\min}$, $d_{\max}$ are the minimum and maximum distances within the mini-batch. This assigned weight $w$ dynamically changes according to the sample statistics in the mini-batch, enabling adaptive attention per iteration.

To address the distinction between positive and negative sample weighting, we introduce separate weighting terms for positive and negative pairs. Specifically, $w_{pos}$ corresponds to the weight assigned to positive pairs, while $w_{neg}$ is used for positive-negative pairs.

**Contrastive loss with weighting:** We finally formulate contrastive learning with the proposed weighting scheme. We assume that a mini-batch contains 2N samples in total, including N query samples and their corresponding N positive samples. We introduce separate weighting terms for positives $(I, I^+)$ and negatives $(I, I^-)$ as $w_{pos}$ and $w_{neg}$, respectively. With these weights, our constrastive learning loss based on the NT-Xent loss (Chen et al., 2020) is formulated as:

$$\mathcal{L}_i = -\log \frac{\exp\left(w_{pos} \cdot \mathrm{sim}(\tilde{\mathbf{z}}_i, \tilde{\mathbf{z}}_i^+)/\tau\right)}{\sum_{k=1}^{2N} \mathbb{1}_{[k \neq i]} \exp\left(w_{neg} \cdot \mathrm{sim}(\tilde{\mathbf{z}}_i, \tilde{\mathbf{z}}_k^-)/\tau\right)} \tag{3}$$

Here $\tau$ is a temperature parameter, $\text{sim}(\mathbf{z}, \bar{\mathbf{z}}) = \frac{\mathbf{z}^T \bar{\mathbf{z}}}{\|\mathbf{z}\| \|\bar{\mathbf{z}}\|}$ is the cosine similarity function. Overall, our adaptive weighting enables considering the importance separately for positive and negative samples, while closer samples are assigned with higher weights and more distant ones receive lower weights.

# 4 EXPERIMENTS

In this section, we compare our method with existing baselines for pre-training of the 3D hand pose estimation and conduct ablation experiments to support the validity of our approach. We begin by providing a detailed explanation of the dataset and experimental setup (Sec. 4.1). Next, we demonstrate that our model achieves competitive performance compared with existing methods (Sec. 4.2). Following this, we present the results of ablation studies on weighting design in the pre-training phase (Sec. 4.3). Finally, visualizations are used to illustrate the superiority and efficiency of our approach (Sec. 4.4).

## 4.1 EXPERIMENTAL SETUP

**Pre-training datasets:** We curate a large collection of hand images from two major video datasets, Ego4D (Grauman et al., 2022) and 100DOH (Shan et al., 2020), featuring egocentric and exocentric views respectively. From Ego4D, a vast egocentric video dataset with 3,670 hours of footage, we extracted 1.0M hand images from 8K videos. Similarly, from the exocentric dataset 100DOH, which includes 131 days of YouTube footage, we extract 1.0M hand images from 20K videos. These extensive datasets provide diverse hand-object interactions across different views. We also prepare pre-training data with varying amount. "Exo-X" and "Ego-X" denote 100DOH and Ego4D datasets with X images selected randomly (*e.g.*, X = 50K, 100K, ..., 1M, 2M). "Ego&Exo-2M" shows our final set combining both datasets with full images (*i.e.*, 1.0M for each).

**Fine-tuning datasets:** We conduct fine-tuning experiments on three datasets with 3D hand pose ground truth in various data size and viewpoints: exocentric datasets from FreiHand (Zimmermann et al., 2019) and DexYCB (Chao et al., 2021), and an egocentric dataset AssemblyHands (Ohkawa et al., 2023b). FreiHand consists of 130.2K training frames and 3.9K test frames, with both green screen and real-world backgrounds. DexYCB contains 325.3K training images and 98.2K test images, focusing on natural hand-object interactions. AssemblyHands, the largest of the three, includes 704.0K training samples and 109.8K test samples, collected in object assembly scenarios. Following (Spurr et al., 2021), we prepare 10% of the labeled FreiHand dataset, which is denoted as "FreiHand*", especially used for ablation studies. This allow us to assess the performance in a limited supervision setting.

**Implementation details:** For similar hands mining, we choose the PCA embedding size as $D = 14$. For the pre-training framework, we use ResNet-50 (He et al., 2016) as the encoder. Throughout the pre-training phase, all models are trained using LARS (You et al., 2017) with ADAM (Kingma & Ba, 2014) optimizer, with the learning rate of 3.2e-3. Following (Spurr et al., 2021), SimCLR employs scale and color jitter as image augmentation, while PeCLR and SiMHand utilize scale, rotation, translation, and color jitter. We use resized images with $128 \times 128$ as the input. We set the temperature parameter $\tau$ of contrastive learning as 0.5. We use 8 NVIDIA V100 GPUs with a batch size of 8192 for pre-training.

For fine-tuning, we initialize our model with the pre-trained encoder $E(\cdot)$ and then fine-tune with a 3D pose regressor on the labeled datasets. The 3D regressor involves 2D heatmap regression and 3D localization heads, similar to DetNet (Zhou et al., 2020). We use a single NVIDIA V100 GPU with a batch size of 128. We provide more additional details in supplement.

**Evaluation:** We use the following evaluation metrics: the mean per joint position error (MPJPE) in millimeters, which compares model predictions against ground-truth data, and the percentage of correct keypoints based on the area under the curve (PCK-AUC), which measures the proportion of predicted keypoints that fall within a specified distance (20mm to 50mm) from the ground truth with varying thresholds.

Table 1: **Comparison with the state of the art.** We show 3D hand pose estimation accuracy (MPJPE↓) on the FreiHand (Exo) (Zimmermann et al., 2019), DexYCB (Exo) (Chao et al., 2021) and AssemblyHands (Ego) (Ohkawa et al., 2023b) . The best results are highlighted in **bold**, and the second-best results are underlined. SiMHand achieves the best results across various datasets.

| Method | Pre-training | FreiHand (Exo) | | DexYCB (Exo) | | AssemblyHands (Ego) | |
|---|---|---|---|---|---|---|---|
| | | *MPJPE ↓* | *PCK-AUC ↑* | *MPJPE ↓* | *PCK-AUC ↑* | *MPJPE ↓* | *PCK-AUC ↑* |
| w/o pre-training | - | 19.21 | 85.61 | 19.36 | 84.80 | 19.17 | 85.61 |
| SimCLR | Exo-1M | 19.30 | 85.36 | 20.13 | 83.75 | 20.01 | 84.21 |
| | Ego-1M | 19.36 | 85.09 | 20.22 | 83.50 | 20.32 | 83.85 |
| | Ego&Exo-2M | 20.07 | 84.32 | 21.09 | 82.25 | 21.24 | 82.29 |
| PeCLR | Exo-1M | 19.58 | 84.71 | 18.39 | 86.33 | 19.12 | 85.64 |
| | Ego-1M | 19.07 | 85.62 | 18.99 | 85.40 | 19.20 | 85.57 |
| | Ego&Exo-2M | 18.19 | 86.76 | 18.06 | 86.82 | 18.88 | 86.03 |
| **SiMHand (Ours)** | Exo-1M | 16.73 | 88.66 | 17.34 | 87.84 | 18.50 | 86.56 |
| | Ego-1M | 16.15 | 89.48 | 16.99 | 88.34 | 18.26 | 86.95 |
| | **Ego&Exo-2M** | **15.79** | **90.04** | **16.71** | **88.86** | **18.23** | **86.90** |

| Method | Pre-training size | FreiHand* | |
|---|---|---|---|
| | | *MPJPE ↓* | *PCK-AUC ↑* |
| w/o pre-training | - | 48.19 | 49.17 |
| SimCLR | Ego-50K | 53.94 | 42.54 |
| PeCLR | | 47.42 | 49.85 |
| SiMHand | | **35.32** | **63.35** |
| SimCLR | Ego-100K | 53.49 | 43.12 |
| PeCLR | | 46.00 | 51.50 |
| SiMHand | | **31.06** | **68.66** |
| SimCLR | Ego-500K | 49.91 | 47.61 |
| PeCLR | | 43.18 | 54.15 |
| SiMHand | | **28.27** | **72.97** |
| SimCLR | Ego-1M | 46.17 | 50.62 |
| PeCLR | | 34.42 | 64.93 |
| SiMHand | | **23.68** | **79.62** |

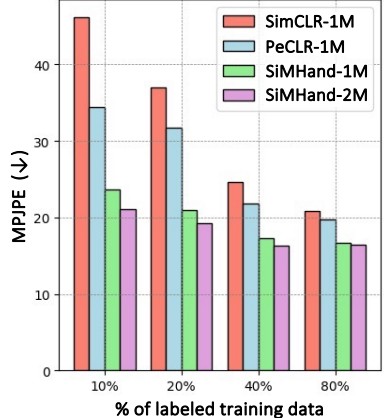

Table 2: **Comparison with different pre-training data sizes.** '*' indicates that we use a small amount of training data for fine-tuning to validate the effectiveness of the pre-trained model. Our method demonstrates a leading advantage across all pre-training data scales.

Figure 4: **Comparison with different data availability in fine-tuning on Frei-Hand.** Variations in the percentage of labeled data correspond to different subsets of the fine-tuning dataset, following the experimental design in (Spurr et al., 2021).

## 4.2 MAIN RESULTS

We compare our method with previous works for 3D hand pose estimation (Tab. 1). To make a fair comparison, we evaluate all pre-training datasets of the same size against previous methods.

**Comparison to contrastive learning methods:** We compare our pre-training method with previous methods (Chen et al., 2020; Spurr et al., 2021) in 3D hand pose estimation (Tab. 1). We observe that our method significantly outperforms SimCLR and PeCLR across various datasets under the equal pre-training data setups. When we compare our method against a randomly initialized model (w/o pre-training), SiMHand improves performance by 17.7% over the scratch baseline.

In more details, our approach achieves a 15.31% improvement over previous methods PeCLR with Ego-1M pre-training on the FreiHand. We observe that SimCLR shows limited performance compared to the random initialization. This suggests pre-training without geometric prior (*i.e.*, without geometric augmentation) does not always help hand pose estimation, requiring spatial keypoint regression. In contrast, our method demonstrates significant performance gain on larger datasets, with a 10.53% gain on DexYCB and a 4.90% improvement on AssemblyHands compared to PeCLR.

Table 3: **Ablation study of proposed modules.** We compare with and without our proposed modules in different methods. The experimental results demonstrate the generality of our method.

| Method | Proposals | | FreiHand* | |
|---|---|---|---|---|
| (Pre-training size) | **Similar hands** | **Adaptive weighting** | *MPJPE ↓* | *PCK-AUC ↑* |
| SimCLR | × | × | 53.49 | 43.12 |
| (Ego-100K) | × | ✓ | **52.58 (1.8% ↓)** | **44.70 (1.58% ↑)** |
| PeCLR | × | × | 46.00 | 51.50 |
| (Ego-100K) | × | ✓ | **44.61 (3.0% ↓)** | **53.37 (1.87% ↑)** |
| **SiMHand** | ✓ | × | 31.06 | 68.66 |
| **(Ego-100K)** | ✓ | ✓ | **28.84 (7.18% ↓)** | **71.07 (2.41% ↑)** |

These results confirm that our model consistently achieves superior performance across various fine-tuning datasets.

Furthermore, we pre-train all methods on the joint pre-training datasets (Ego&Exo-2M). Our approach further improves over the state-of-the-art method (PeCLR), achieving improvements of 13.19%, 7.4%, and 3.4% on the FreiHand, DexYCB, and AssemblyHands, respectively. Compared to the pre-training with 1M samples (Ego-1M), doubling the pretraining data with Ego&Exo-2M results in a 2.28% improvement on the FreiHand dataset. Notably, our method shows particular strength in effectively handling larger, more varied datasets. This robust performance demonstrates that our approach is highly effective and reliable for hand pose pre-training.

**Ego & Exo view analysis:** We evaluate how pre-training with egocentric views (Ego4D) and exocentric views (100DOH) affects the performance in datasets with their corresponding views, namely AssemblyHands for egocentric and FreiHand and DexYCB for exocentric views. Interestingly, matching pre-training viewpoints does not consistently enhance performance, indicating that the view gaps have limited effects. Instead, factors like dataset diversity and the characteristics of pre-training methods are more crucial in boosting performance. Combining the two datasets (the last row of Tab. 1) leads to the best performance in all three datasets, underscoring the potential of enriching data diversity with various camera views.

## 4.3 ABLATION EXPERIMENTS

This section presents ablation studies on SiMHand, focusing on four aspects: 1) pre-training dataset size, 2) fine-tuning dataset size, 3) adaptive weighting, and 4) Top-K similar hands. First, we examine the size of the pre-training dataset using various methods, showing that our approach maintains superior performance across different sizes (Tab. 2). Second, inspired by (Zimmermann et al., 2019), we explore fine-tuning dataset size, demonstrating significant gains even with limited data (Fig. 4). Furthermore, we also highlight the adaptive weighting design, which consistently outperforms comparison methods (Tab. 3). Finally, we conduct ablation analysis according to different levels of similarity in the assigned positive hand pairs. (Tab. 4).

**Effect of pre-training data size:** We study results with different sizes of pre-training data, namely 50K, 100K, 500K, and 1M in Tab. 2. The results demonstrate that SiMHand reliably outperforms the other methods across all settings, with improvement as the pre-training data size increases. With changes in the size of the pre-training data from 50K to 1M, SiMHand achieves a reduction in MPJPE from 35.32 to 23.68. The useful insights we can gather from this table include: 1) The SiMHand method holds a leading advantage across various pre-training size. 2) As the size of the pre-training dataset increases, the improvement for fine-tuning with limited labels is substantial.

**Effect of fine-tuning data size:** Fig. 4 illustrates the experiment under different proportions of labeled fine-tuning data, namely 10%, 20%, 40%, and 80% in FreiHand. Note that we denote methods with "-1M/2M" as those pre-trained on the Ego-1M and the Ego&Exo-2M sets, respectively. The results show that SiMHand-1M brings error reduction, achieving remarkably lower MPJPE scores with merely 10% of labeled data. SiMHand-1M delivers the best performance over different size of fine-tuning data, compared to SimCLR-1M and PeCLR-1M. SiMHand-2M further shows improvement over SiMHand-1M, while the gains become marginal as labeled data increase. From

Table 4: **Pre-training performance at different similarity ranks (Top-K).** It can be seen that as the similarity rank increases, the pre-training performance deteriorates.

| Method (Pre-training size) | Top-K | FreiHand* | |
|---|---|---|---|
| | | *MPJPE ↓* | *PCK-AUC ↑* |
| | **Top-1** | **31.06** | **68.66** |
| | Top-2 | 31.46 | 67.89 |
| | Top-5 | 31.85 | 67.20 |
| | Top-10 | 31.87 | 67.18 |
| **SiMHand** | Top-50 | 31.53 | 67.59 |
| **(Ego-100K)** | Top-100 | 31.54 | 67.70 |
| | Top-500 | 32.61 | 66.76 |
| | Top-1000 | 34.05 | 65.14 |
| | Top-5000 | 35.34 | 62.79 |

this analysis, we can draw two key conclusions: 1) The improvement resulting from an increase of pre-training data becomes less significant as the amount of fine-tuning data increases; 2) SiM-Hand maintains a strong advantage in scenarios with limited labeled data, particularly when larger pre-training data are used.

**Effect of adaptive weighting:** We validate the proposed adaptive weighting and its generality when applied to the other methods in Tab. 3. On the Ego-100K pre-training set, the MPJPE scores after adaptive weighting decrease by 1.8% and 3.0% for SimCLR and PeCLR, respectively, while PCK-AUC increases by 1.58% and 1.87%. This indicates that the proposed weighting excels in its applicability to various pre-training methods. In our SiMHand method, applying adaptive weighting reduces MPJPE from 31.06 to 28.84, a 7.18% decrease, while PCK-AUC improves from 68.66 to 71.07, a 2.41% increase. We find the effectiveness of the proposed weighting when combined with the mined similar hands.

**Learning from Top-K similar hands:** We test pre-training with different similarity levels of positive samples in Tab. 4. As illustrated in Fig. 2, we can sample similar pairs according to the distance ranking (*e.g.*, K = 1, 2, ..., 5000), where Top-1 is used to produce our final results. The performance trend is initially subtle and somewhat fluctuating (Top-1∼100) but becomes increasingly pronounced after Top-100. This indicates that as the similarity between positive samples increases, the global trend decreases accordingly. Notably, using Top-5000 similar hand samples as positive samples decreases the MPJPE by 13.78% compared to Top-1. This study provides two insights: 1) Similar samples with subtle noisiness (*e.g.*, 1∼100) exhibit minimal variation in performance, indicating that slight differences in similarity within this range do not significantly impact the pre-training outcome. This suggests that the model is robust to minor variations when the positive samples are highly similar. 2) The results support the validity of using Top-1 positive samples to produce final results, as they consistently exhibit the best performance. This highlights the importance of selecting the most similar samples in contrastive learning.

## 4.4 VISUALIZATION

In this section, we compare the fine-tuning results of various pre-training methods through detailed visualizations on different datasets (Fig. 5). The pre-training model is trained on the Ego&Exo-2M dataset and fine-tuned on the FreiHands (Zimmermann et al., 2019) and DexYCB (Chao et al., 2021) datasets, respectively. We provide additional visualization in the supplementary material.

From the left four columns of Fig. 5, the visualization results show that SiMHand performs better in pose estimation, with results closer to the ground truth, compared to the other methods in Frei-Hands (Zimmermann et al., 2019) dataset. In particular, SiMHand outperforms the other methods in challenging environments, such as those with varying lighting conditions, by better capturing hand poses. These visual outputs highlight its robustness across various scenarios, solidifying its potential for real-world applications.

As shown in the right four columns of Fig. 5, we highlight the occluded regions in the original images of DexYCB (Chao et al., 2021) dataset using red circles. The results show that SiMHand is more effective in tackling occlusion problems. Our pre-training method effectively addresses partially

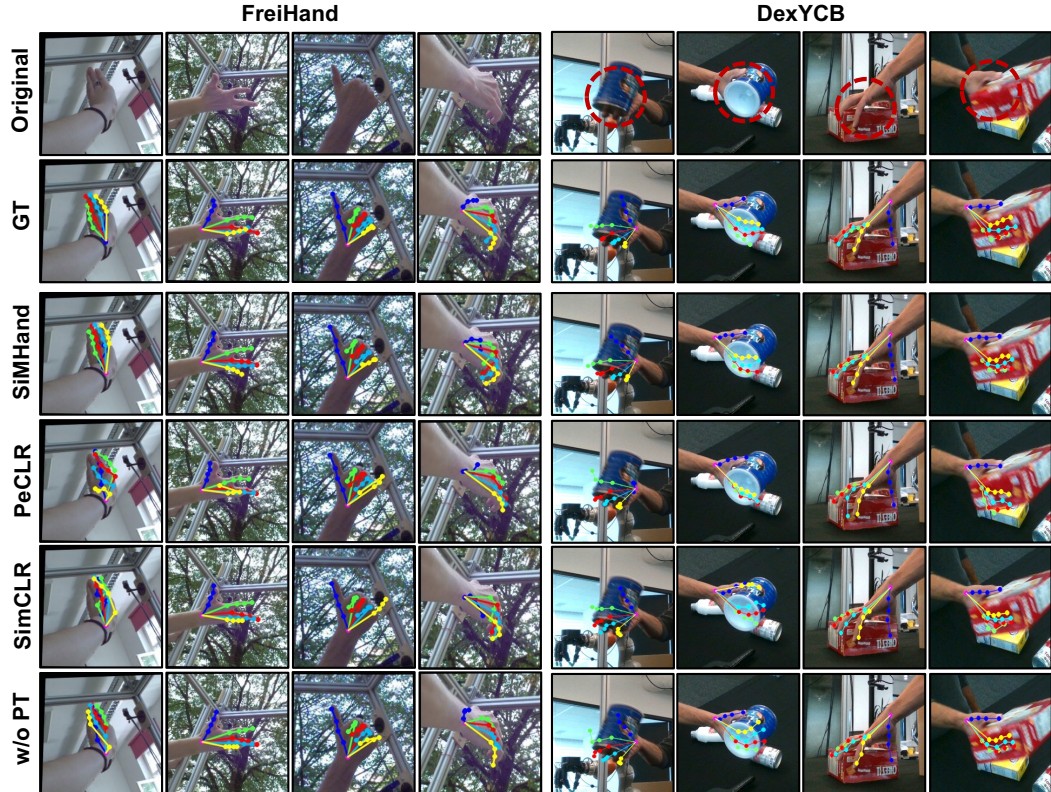

Figure 5: **Visualization of FreiHand (Zimmermann et al., 2019) and DexYCB (Chao et al., 2021).** The first four columns on the left display the results for FreiHand, while the last four columns on the right show the results for DexYCB (GT: Ground Truth; PT: Pre-training). It can be observed that SiMHand pre-training method achieves better results.

occluded images by utilizing similar, though not identical, hand images, where the occluded parts in the query image may be visible in the corresponding similar hand image, and vice versa.

## 5 CONCLUSION

We introduce SiMHand, a contrastive learning framework for pre-training 3D hand pose estimators by mining similar hand pairs from large-scale in-the-wild images. Our approach leverages similar hand pairs from diverse videos, significantly enhancing the information gained during pre-training compared with existing methods. Experiments show that our pre-training method achieves competitive performance in 3D hand pose estimation across multiple datasets, outperforming previous pre-training approaches and demonstrating the benefits of large-scale pre-training with in-the-wild images. We hope this work can lay a foundation for future research on pre-training of 3D hand pose estimation.

ACKNOWLEDGMENTS

This work was supported by the JST ACT-X Grant Number JPMJAX2007, JST ASPIRE Grant Number JPMJAP2303, JSPS KAKENHI Grant Number JP24K02956, JP22KF0119, and NSFC Grant Number 62376090.

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

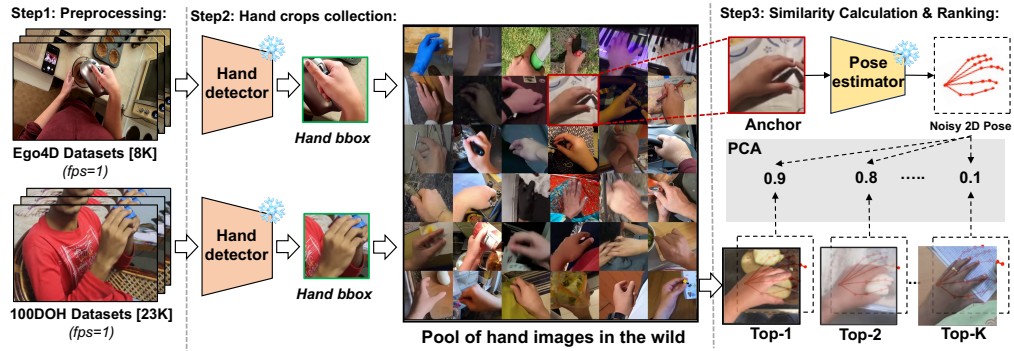

Figure 6: **Overview of data preprocessing and similar hands mining.** This image illustrates a three-step process for SiMHand pre-training using datasets from Ego4D and 100DOH. **Step 1** involves preprocessing the datasets to extract relevant frames. **Step 2** employs a hand detector to crop hand regions from these frames, creating a diverse pool of hand images in the wild. **Step 3** calculates similarity and ranks the images using a pose estimator and PCA, producing a sorted list of hand poses, from the most similar to the least similar to a given anchor pose.

## 6 APPENDIX

### 6.1 CONSTRUCTION OF LARGE-SCALE IN-THE-WILD HAND DATABASE

This section presents our method for constructing a large-scale hand image dataset by extracting and processing hand images from various video datasets. We outline key preprocessing steps, including *1) preprocessing*, *2) hand region detection*, and *3) similarity calculation & ranking*.

**Preprocessing:** We prepare two large-scale video datasets: Ego4D, containing 8k frames, and 100DOH, with 23k frames, both sampled at 1 *fps*. As shown in Fig. 6, first-person and third-person hand images exhibit significant differences.

**Hand region detection:** After extracting frames from Ego4D and 100DOH, we use a lightweight, fixed-weight network to detect hand regions via bounding boxes. Specifically, we adopt the method from (Shan et al., 2020) and store all detected bounding boxes in sequence. This step constructs a large-scale hand image dataset as Tango et al. (2022).

**Similarity calculation & ranking:** Once the hand image dataset is built, we use a lightweight, fixed-weight network to extract raw keypoints for each sample via MediaPipe (Lugaresi et al., 2019). To reduce noise, we apply PCA as described in Sec. 3.1. We then compute similarity scores for a given query image $I$ using Eq. 1 and rank the remaining samples accordingly. This process yields a large-scale set of in-the-wild hand images with similar characteristics. For instance, in Ego4D, given a query sample $I$, we retrieve all similar hand images and construct a ranked sequence, referred to as "Top-K". The Top-1 image in this sequence serves as the positive sample $I^+$ for contrastive learning, enhancing the effectiveness of SiMHand pre-training. As shown in Tab. 4, our experiments validate that selecting Top-1 as the positive sample $I^+$ is the optimal strategy.

### 6.2 FINETUNE FOR 3D HAND POSE ESTIMATION

In the fine-tuning stage, we discard the projection head and fine-tuning only the encoders. We load the pre-training model weights into a heatmap-based 3D hand pose estimationand predition method: DetNet(Zhou et al., 2020). To train DetNet, we utilize a comprehensive loss function designed to optimize both 2D pose estimation and 3D spatial localization. The loss function is defined as:

$$\mathcal{L}_{\text{heat}} + \mathcal{L}_{\text{loc}} + \mathcal{L}_{\text{delta}} + \mathcal{L}_{\text{reg}} \tag{4}$$

where $\mathcal{L}_{\text{heat}}$ ensures that the predicted heatmaps $H$ align closely with the ground truth heatmaps $H^{\text{GT}}$, $\mathcal{L}_{\text{loc}}$ and $\mathcal{L}_{\text{delta}}$ measure the discrepancies between the predicted location maps $L$ and delta maps $D$ and their corresponding ground truth $L^{\text{GT}}$ and $D^{\text{GT}}$, with $H^{\text{GT}}$ weighting these discrepancies to focus on the maxima of the heatmaps. Additionally, $\mathcal{L}_{\text{reg}}$ is an $L2$ regularization term to

| Method | Pre-training size | FreiHand* | |
|---|---|---|---|
| | | *MPJPE* ↓ | *PCK-AUC* ↑ |
| PeCLR | | 47.42 | 49.85 |
| TempCLR | Ego-50K | 45.17 | 52.40 |
| SiMHand | | **35.32** | **63.35** |
| PeCLR | | 46.00 | 51.50 |
| TempCLR | Ego-100K | 44.54 | 53.28 |
| SiMHand | | **31.06** | **68.66** |

Table 5: **Comparison with the TempCLR method.**'*' indicates that we use a small amount of training data for fine-tuning to validate the effectiveness of the pre-trained model. TempCLR outperforms PeCLR by a modest margin, whereas SiMHand achieves a significant performance improvement over TempCLR.

prevent overfitting. Note that after passing through the encoder, we made simple adjustments to the model, applying some upsampling to the features to fit the input.

This multi-task learning framework enables the network to simultaneously learn pose features from 2D images and spatial information from 3D data, enhancing the accuracy and robustness of detection in real-world applications. For more details on fine-tuning, please refer to the (Zhou et al., 2020).

### 6.3 COMPARISON WITH TEMPCLR METHOD

We conduct an experimental comparison with the TempCLR (Ziani et al., 2022) method. TempCLR proposes a pre-training framework for 3D hand reconstruction using time-coherent contrastive learning and demonstrates better performance compared to PeCLR (Spurr et al., 2021). Although TempCLR primarily focuses on reconstruction tasks, the parametric model it uses can also output 3D pose results, making it valuable to further compare our method with TempCLR.

However, TempCLR has certain limitations in data collection and the effectiveness of contrastive learning. First, TempCLR treats hands from adjacent frames as positive samples during training. In dynamic egocentric videos, hand occlusions or detection failures often lead to missed hand crops in neighboring frames. In addition, images from adjacent frames typically lack background diversity, limiting the contribution of positive sample pairs formed from neighboring frames in contrastive learning.

In contrast to TempCLR, our method, SiMHand, significantly improves performance. SiMHand leverages similar hand images, which provide richer diversity in features, including various types of hand-object interactions, diverse backgrounds, and varying appearances. These features allow SiMHand to effectively increase the diversity of positive samples in contrastive learning, resulting in superior pre-training performance.

We further validate our approach on two different size of pre-training data, consisting of 50K and 100K hand images from the Ego4D dataset (Grauman et al., 2022). Tab. 5 shows the significant progress made by SiMHand compared to TempCLR and PeCLR.

From the experimental results, TempCLR demonstrates better performance than PeCLR, which matches the conclusion of the original paper. However, SiMHand provides more valuable positive samples for contrastive learning, leading to better results during the fine-tuning phase of 3D hand pose estimation tasks.

### 6.4 COMPARISON WITH WEAKLY-SUPERVISED LEARNING SETTING

We compare our method with a weakly-supervised learning setting that uses 2D noisy keypoints assigned on in-the-wild images. In a weakly-supervised learning setting, noisy 2D keypoints are directly used as supervision signals during network training. The model treats these 2D keypoints as targets, computing the loss between the predicted keypoints and the provided 2D keypoints (*e.g.*, heatmap-based loss). However, our experiments reveal that directly conducting joint training on labeled and unlabeled data results in degraded performance due to the noise and unreliability of

| Setting | Unlabeled data | FreiHand* | |
|---|---|---|---|
| | | *MPJPE* ↓ | *PCK-AUC* ↑ |
| Weakly-supervised | Ego-100K | 61.65 | 33.92 |
| Pre-training & Fine-tuning | Ego-100K | **31.06** | **68.66** |

Table 6: **Comparison with weakly-supervised learning setting.** We observe that directly incorporating noisy labels into the joint training in the weakly-supervised setting leads to a decline in model performance, indicating that applying noisy labels for training presents certain challenges.

| Method | Backbone | DexYCB |
|---|---|---|
| | | *MPJPE* ↓ |
| Xiong et al. (2019) | ResNet50 | 25.57 |
| Spurr et al. (2020) | ResNet50 | 22.71 |
| Spurr et al. (2020) | HRNet32 | 22.26 |
| Tse et al. (2022) | ResNet18 | 21.22 |
| Zhou et al. (2020) | ResNet50 | 19.36 |
| SiMHand | ResNet50 | **16.71** |

Table 7: **Comparison of 3D hand pose estimation methods on DexYCB (Chao et al., 2021).**

| Method | Backbone | AssemblyHands |
|---|---|---|
| | | *MPJPE* ↓ |
| Han et al. (2022) | ResNet50 | 32.91 |
| Ohkawa et al. (2023b) | ResNet50 | 21.92 |
| Zhou et al. (2020) | ResNet50 | 19.17 |
| SiMHand | ResNet50 | **18.23** |

Table 8: **Comparison of 3D hand pose estimation methods on AssemblyHands (Ohkawa et al., 2023b).**

the 2D keypoints. As shown in Tab. 6, without any keypoint filtering or correction, the weakly-supervised method performs significantly worse than our pre-training setting.

These findings demonstrate that directly incorporating noisy 2D annotations during weakly-supervised training negatively impacts model performance, particularly when the labels contain high levels of noise.

Before designing our pre-training approach, we identified several limitations of the weakly-supervised setting for large-scale, in-the-wild hand data based on prior experience: (1) *Data scale constraints:* When the amount of noisy hand data is significantly smaller than the noise-free hand training dataset, it may provide some improvement but it is hard to guarantee that such noisy labels are less in larger datasets (e.g., the two million in-the-wild hand images in this study) and (2) *Training efficiency issues*: Introducing large-scale noisy data significantly prolongs training time and slows convergence. In contrast, our pre-training method benefits from such large unlabeled hand images with certain noisiness. This highlights our superiority in exploiting pre-training over the weakly-supervised setting.

## 6.5 COMPARISON WITH THE OTHER 3D HAND POSE ESTIMATION METHODS

To better assess the value of this work and its position within the broader context, we have included comparisons with other related works in the field of 3D hand pose estimation in this section.

As shown in Tab. 7 and 8, the comparative results on the DexYCB (Chao et al., 2021) and AssemblyHands (Ohkawa et al., 2023b) datasets further validate the superiority of our approach across multiple standard datasets, demonstrating the effectiveness of our pretraining strategy and its broad potential for real-world applications.

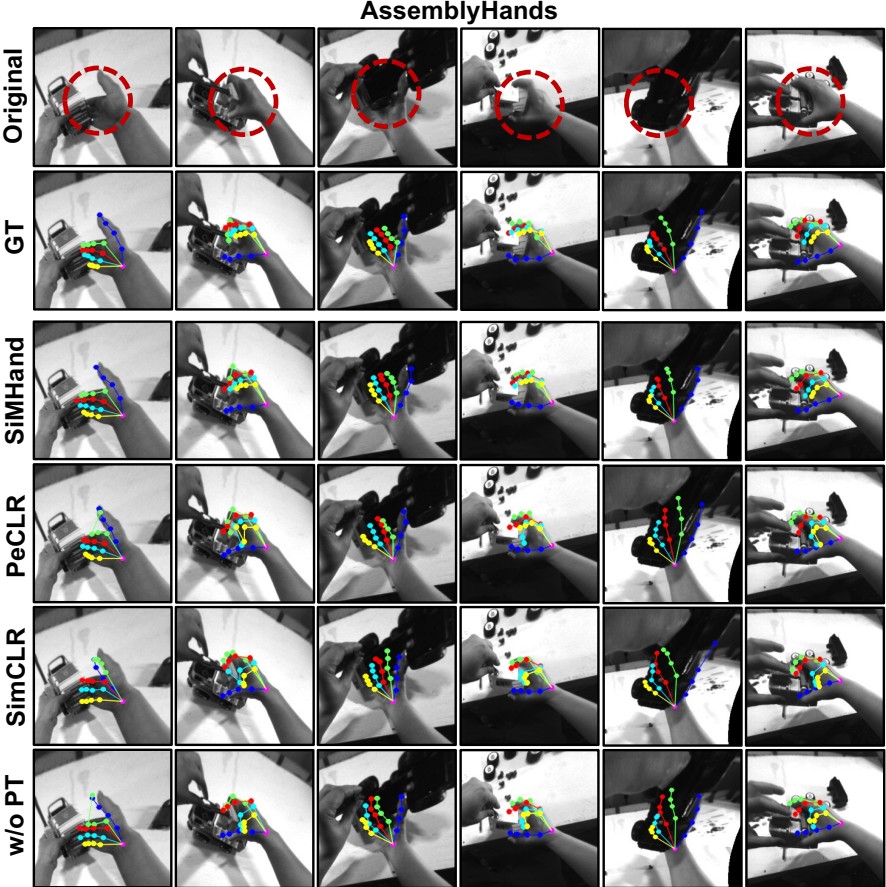

Figure 7: **Visualization of Hand Pose Estimation Results on AssemblyHands.** AssemblyHands Ohkawa et al. (2023b) is a hand pose dataset captured from a first-person perspective during toy assembly. It can be observed that SiMHand pre-training method achieves better results (GT: Ground Truth; PT: Pre-training).

### 6.6 VISUALIZATION OF HAND POSE ESTIMATION RESULTS ON ASSEMBLYHANDS

We show the visualization results of hand pose estimation on another dataset, AssemblyHands Ohkawa et al. (2023b). We highlight instances of hand-object occlusion in the data using red circles. As observed with DexYCB (Chao et al., 2021), SiMHand pre-trained model demonstrates superior performance in handling occlusion during the fine-tuning stage compared to the other pre-training methods, showcasing stronger robustness.

### 6.7 VISUALIZATION OF SIMILAR HANDS

We present the visualization of Top-K similar hand images used to create positive pairs. As shown in Fig. 8, we visualize a set of Top-K similar hand images. The figure displays the query image alongside its corresponding similar hand sequence (Top-K). At the top of Fig. 8, a timeline indicates that the images are deliberately sampled from consecutive frames of the same video.

From these visualizations, we derive three key insights: 1) Using adjacent frames from the same video as positive samples in pre-training lacks diversity, as substantial variations may still exist between samples. 2) As the ranking increases, the similarity between hand images decreases significantly, leading to greater differences that may result in inaccurate feature representations during pre-training. 3) Therefore, selecting the Top-1 image is a proper design to assign diverse yet similar positive samples for the query images.

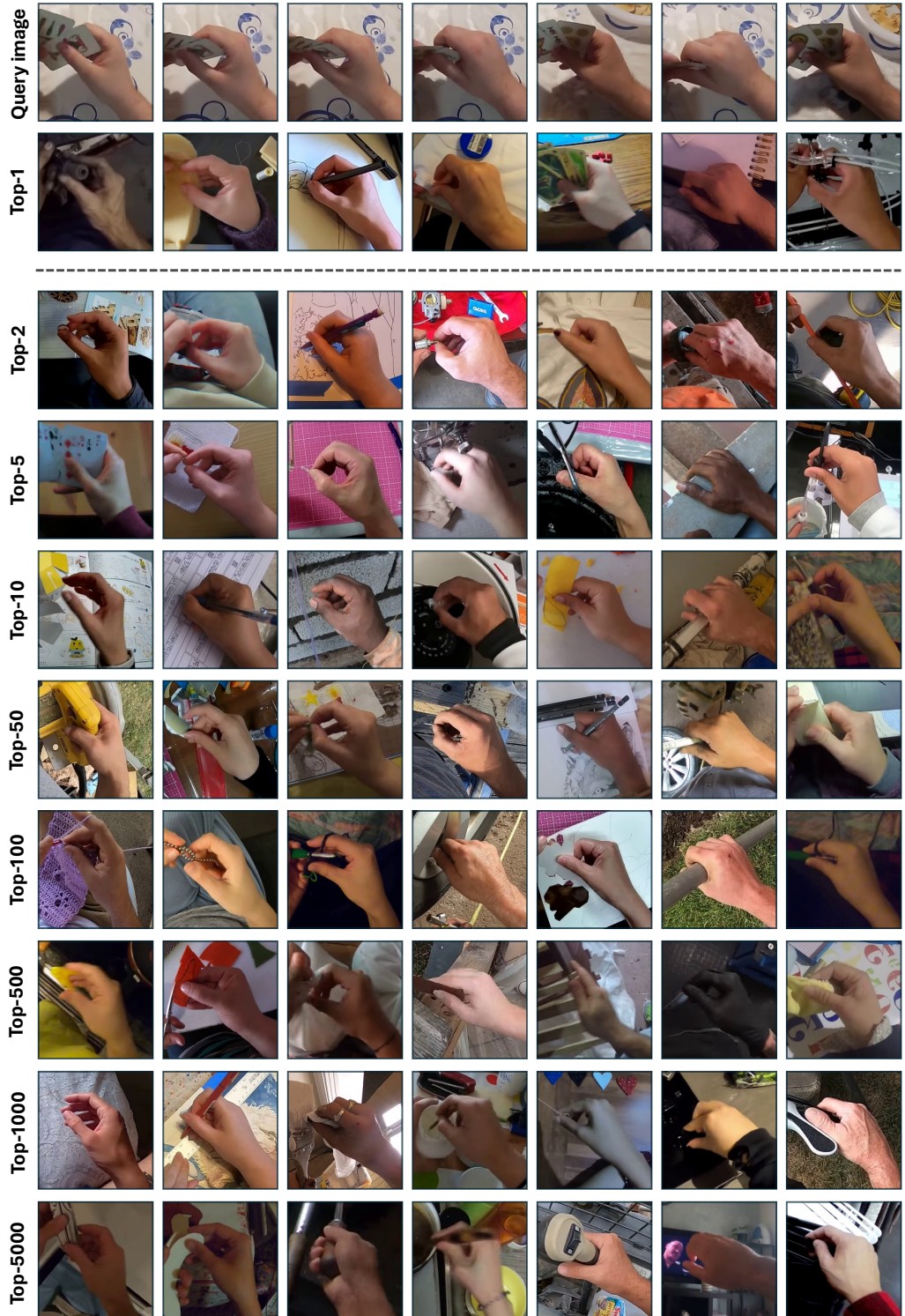

Figure 8: **Visualization of similar hand samples in Top-K.** As the ranking increases, the differences between hand samples become more pronounced.

