# OpenReview forum: "SiMHand: Mining Similar Hands for Large-Scale 3D Hand Pose Pre-training"
_ICLR.cc/2025/Conference — ICLR 2025 Poster_

### Official Review · Reviewer_PnQd · 2024-10-31

**Soundness:** 3
**Presentation:** 2
**Contribution:** 2
**Rating:** 8
**Confidence:** 5

**Summary:**

This paper addresses the task of self-supervised learning for 3D hand pose estimation from monocular RGB. The authors build on prior work in the area and improve upon it in three main areas: 1) Use of noisy 2D supervision to mine positive samples 2) Adaptive weighting that weighs positive and negative samples based off of their distance of the 2D keypoints 3) Processing and use of Ego4D and 100DOH for self-supervision.
Their proposed method first constructs a pose embedding based off of the noisy acquired 2D keypoints using an off-the-shelf predictor. This pose embedding is then used to mine positive samples given a query image.
These positive samples are used within the contrastive loss as positive samples, whereas the remaining image in the batch is marked as negative. The positive and negative samples are weighted additionally using weights that are computed based off of the scaled euclidean distance of their 2D keypoints.
The self-supervised model is trained on Ego4D and/or 100DOH. Those datasets have been processed using an off-the-shelf hand detector model. Supervised training was done on a variety of supervised datasets. Experimental results show large improvement across all benchmark datasets compared to prior self-supervised models.

**Strengths:**

- The paper empirically verifies the improvement over prior self-supervised models. Self-supervision is a rather underexplored area in hand pose estimation and can lead to potentially great benefit as foundation models.
- The improvements are substantial
- The paper is easy to understand

**Weaknesses:**

- The method shows great improvement over prior self-supervised methods through the use of noisy 2D annotations. However, its use is a rather involved process: it first needs to be embedded, then used during pre-training before then performing supervised fine-tuning. Instead, why not just use the noisy 2D annotations directly as a form of weak-supervision? In fact, this has been done in prior work [1] and has lead to substantial improvements. In order to properly verify the usefulness of the authors proposed method, there first needs to be a baseline showcasing that the straightforward addition of the noisy 2D annotation during pre-training or supervised training performs comparatively worse. Otherwise why should one employ the authors proposed method? Due to my own experiences in the field, I fear that the weak-supervision approach will outperform the authors proposed approach.
- The paper does not compare to other related work in the field for which test results on FreiHand, DexYCB and AssemblyHands are available. Without these, we cannot assess properly the value of this work and how it fits in overall.


[1] Weakly-Supervised Mesh-Convolutional Hand Reconstruction in the Wild, Kulon et al., CVPR'20

**Questions:**

- How does this work compare to a weakly-supervised approach with noisy annotations?
- How does this work perform compared to other related work on the tested datasets?
- Instead of using weights, could one instead not use a more appropriate loss that will automatically lead to larger effects depending on the samples weights? For example, MSE will automatically weight the contributions of more distant samples stronger.
- L143-144: Why balance the number of left and right hand if they all end up being converted to right-handed images?
- Eq 1: Why not use the cosine similarity which is more popular for distances in feature space?
- Fig3: The colored boxes at the end of the model pipeline seems to be in the wrong order. E.g the figure shows positive samples minimizing alignment.
-L238-239: rough -> noisy
- Table 1: What is "baseline"? This needs to be explained in the image caption
- Table 1: Why are the worst result of SimCLR in bold? Shouldn't the most performant number be in bold?
- Not all figures and tables are referred to in text.
- Table 3: inconsistent capitalization of simclr etc. This also occurs occasionally in the text.

---

> ### Author Response · Authors · 2024-11-23
> **Response to Reviewer PnQd  (R3).**
>
> We deeply appreciate the time and effort reviewer PnQd (R3) took to review our paper, **and especially the valuable suggestion to compare with the weakly-supervised approach via training of noisy annotations.** Below are our responses to all the questions (W: Weakness; Q: Question; A: Answer):
>
> >**Q1 (W1)**: How does this work compare to a weakly-supervised approach with noisy annotations?
>
> **A1**: We appreciate the reviewer’s recommendation for another comparison method to improve our work, ie., a weakly-supervised baseline using 2D noisy keypoints. **We have added a comparison with the experimental results of a weakly-supervised setting.** We find that a naive joint training across the labeled and unlabeled data rather worsens the performance due to the noisiness and unreliability of the 2D keypoints. We believe that additional keypoints filtering (to remove highly noisy labels) or a scheme that corrects 2D keypoints would be necessary to effectively utilize the noisy labels on unlabeled data. Notably, our pre-training method has superiority as it performs well without such additional filtering and keypoint correction methods. Based on the same experimental setup, the results on FreiHand* for the two different settings are as follows:
>
> | Setting | Unlabeled data | MPJPE (↓) | PCK-AUC (↑) |
> |---------|---------|---------|---------|
> | Weakly-supervised | Ego4D-100K | 61.65   | 33.92   |
> | Pre-training & Fine-tuning | Ego4D-100K | 31.06 |  68.66 |
>
> As reviewer PnQd mentioned, this baseline demonstrates that the approach of directly adding noisy 2D annotations during pre-training or supervised training results in noticeably worse performance, especially when there are highly noisy labels and significant differences from the officially provided labeled samples, as shown in the example above.
>
> Before designing our pre-training method, we identified the following issues with weakly-supervised setting applied to larger-scale, in-the-wild hand data based on past experience:
> * When the amount of noisy hand data is significantly smaller than the Official training data size, it can provide some improvement in a weakly-supervised setting, but it's hard to apply to large-scale, in-the-wild hand data **(e.g., 2 million in-the-wild hand images in our work)**;
> * Introducing a larger-scale noisy hand data in a weakly-supervised setting extends training time and slows convergence;
> * The weakly-supervised setting lacks cross-dataset generalizability, as a model trained on dataset _A_ using a weakly-supervised setting performs poorly on datasets _B_ or _C_.

---

> ### Author Response · Authors · 2024-11-23
> **Response to Reviewer PnQd (R3).**
>
> >**Q2 (W2)**: How does this work perform compared to other related work on the tested datasets?
>
> **A2**: Since our work primarily focuses on achieving more efficient contrastive learning-based pre-training on large-scale wild hand data, we have dedicated more space and effort to presenting experiments related to the pre-training method. **To address the reviewer's question, we have included more comparative results on the test datasets.** We present some more comparisons below:
>
> **For DexYCB**:
> | Method | Backbone | MPJPE(↓) |
> |---------|---------|---------|
> | A2J [5] | ResNet50 | 25.57 |
> | Spurr et al. [6] | ResNet50 | 22.71 |
> | Spurr et al. [6] | HRNet32 | 22.26 |
> | Tse et al. [7] | ResNet18 | 21.22 |
> | Minimalhands [4] | ResNet50 | 19.36 |
> | **Ours** | **ResNet50** | **16.71** |
>
> **For AssemblyHands**:
> | Method | Backbone | MPJPE(↓) |
> |---------|---------|---------|
> | UmeTrack [8] | ResNet50 | 32.91 |
> | SVEgoNet [9] | ResNet50 | 21.92 |
> | Minimalhands [4] | ResNet50 | 19.17  |
> | **ours** | **ResNet50** | **18.23** |
>
> **Reference**:
> >[5] Fu Xiong et al. “A2J: Anchor-to-joint regression network for 3D articulated pose estimation from a single depth image.”
>
> >[6] Adrian Spurr et al. “Weakly supervised 3D hand pose estimation via biomechanical constraints.”
>
> >[7]  Tze Ho Elden Tse et al. “Collaborative learning for hand and object reconstruction with attention-guided graph convolution.”
>
> >[8] Shangchen Han et al. "UmeTrack: Unified multi-view end-to-end hand tracking for VR"
>
> >[9] Takehiko Ohkawa et al. "AssemblyHands: Towards Egocentric Activity Understanding via 3D Hand Pose Estimation"

---

> ### Author Response · Authors · 2024-11-23
> **Response to Reviewer PnQd (R3).**
>
> >**Q3**: Instead of using weights, could one instead not use a more appropriate loss that will automatically lead to larger effects depending on the samples weights? For example, MSE will automatically weight the contributions of more distant samples stronger.
>
> **A3**: Based on the reviewer’s suggestion, **we implemented MSE as a loss function to minimize the distance between samples.** We pre-train the model on the Ego4D-100K dataset and fine-tune on FreiHAND*. The experimental results below:
>
>  | Setting | MPJPE (↓) | PCK-AUC (↑) |
> |---------|---------|---------|
> | w/o AW | 31.06  | 68.66 |
> | MSE | 49.48  | 47.38 |
> | **w/ AW** | **28.84**  | **71.07** |
>
> (AW: adaptive weighting)
>
> >**Q4**: L143-144: Why balance the number of left and right hand if they all end up being converted to right-handed images?
>
> **A4**: Performing a flip on left-hand images to convert to right-hand images is a standard approach in 3D hand pose estimation. Since our hands are symmetrical, this approach is widely used to reduce the complexity of the input space and make it easier to learn the hand pose. For the downstream use cases, the converted images and predicted poses are flipped again to align with the original images.
>
> >**Q5**: Eq 1: Why not use the cosine similarity which is more popular for distances in feature space?
>
> **A5**: We use the Euclidean distance in Eq 1 because the pose embedding originates from 2D keypoints rather than general image features. Analogy to measuring the distance in pose with the Euclidean metric (e.g., MPJPE in the evaluation of pose estimation), we select the Euclidean metric.
>
> >**Q6**: Fig3: The colored boxes at the end of the model pipeline seem to be in the wrong order. E.g the figure shows positive samples minimizing alignment. -L238-239: rough -> noisy
>
> **A6**: Thank you for pointing out the ordering issue in Figure 3 regarding "Minimize alignment" and "maximizing alignment," which could lead to ambiguity in the original version. This issue has been addressed and clarified in the latest uploaded revision.
>
> >**Q7**: Table 1: What is "baseline"? This needs to be explained in the image caption.
>
> **A7**: The baseline model used in Tab.1 and Tab.2 follows our fine-tuning algorithm of ResNet50 + heatmap regression [4], which are trained from scratch, i.e., the baseline without pre-training. To improve the clarity, we have replaced the term "baseline model" with "w/o pre-training" in the revised manuscript.
>
> >**Q8**: Table 1: Why are the worst results of SimCLR in bold? Shouldn't the most performant number be in bold?
>
> **A8**: Thank you for the reviewer's observation. Based on the reviewer's suggestion, we have adjusted the table formatting in the newly uploaded version.
>
> >**Q9**: Not all figures and tables are referred to in text.
>
> **A9**: In the newly uploaded version, we revise all figures and tables to be properly referenced. Thank you for your suggestion.
>
> >**Q10**: Table 3: inconsistent capitalization of simclr etc. This also occurs occasionally in the text.
>
> **A10**: We have adjusted for case inconsistencies in the newly uploaded version. Thank you.

---

> ### Comment · Reviewer_PnQd · 2024-11-25
> **Response to authors**
>
> I thank the authors for their extensive response to my review. As I believe these results are extremely important to underly the value of the paper, will the authors be including the results as well as the code to these experiments that i requested upon acceptance?

---

> > ### Author Response · Authors · 2024-11-26
> > **Promise to Reviewer PnQd (R3).**
> >
> > Yes. We promise that we will include both the requested results follow the Reviewer PnQd (R3) and the code for the experiments upon acceptance. Thank you for Reviewer PnQd (R3)'s thoughtful and prompt feedback, as well as for recognizing the significance of these results in enhancing our work.

---

> > ### Author Response · Authors · 2024-11-30
> > **Response to Reviewer PnQd (R3).**
> >
> > Dear Reviewer PnQd (R3),
> >
> > We would like to sincerely thank you for reviewer PnQd (R3)'s thoughtful and constructive feedback. We truly appreciate reviewer PnQd (R3)'s recognition of the improvements we’ve made in our work, as well as the value of our work. Your acknowledgment of our efforts is greatly encouraging.
> >
> > As a follow-up, we would like to check if there are any further questions or additional aspects reviewer PnQd (R3) would like to discuss with us? We are fully committed to engaging in any further discussions based on your valuable insights.
> >
> > Thank you again for reviewer PnQd (R3)'s time and valuable comments.
> >
> > Best Regards,
> >
> > All Anonymous Authors

---

> > > ### Comment · Reviewer_PnQd · 2024-12-03
> > >
> > > Dear Authors, you have answered all my questions and I am happy to raise my ratings. I thank you for the time and effort you put into creating these additional experiments.

---

> > > > ### Author Response · Authors · 2024-12-03
> > > > **Response to Reviewer PnQd (R3).**
> > > >
> > > > Dear Reviewer PnQd (R3),
> > > >
> > > > Thank you for your recognition of our paper and for raising your ratings! We also appreciate the valuable suggestions you provided.
> > > >
> > > >
> > > > Best Regards,
> > > >
> > > > All Anonymous Authors

---

### Official Review · Reviewer_4Kod · 2024-11-01

**Soundness:** 3
**Presentation:** 3
**Contribution:** 3
**Rating:** 6
**Confidence:** 5

**Summary:**

- The paper explores pretraining for hand pose estimation using a large number of 2D image samples.
- It introduces HandCLR, a contrastive learning-based method. This method expands the definition of positive and negative samples by using pairs with similar actions from different sources, improving upon previous methods that relied solely on data augmentation.
- The authors collected extensive pretraining datasets from 100DOH and Ego4D and studied effective methods for mining similar hand samples.
- They designed a Top-K sampling strategy for positive and negative samples and implemented adaptive weighting.
- Experiments show that the proposed method outperforms baselines in both pretraining and downstream finetuning tasks.

**Strengths:**

- The paper is well-written and easy to follow.
- The motivation behind the proposed method is sound, with comprehensive details from data preparation to training.
- The design of contrastive loss with weighting provides better gradient guidance for samples with different sources and similarities, which is both reasonable and effective.
- The numerous experiments reflect significant effort by the authors.
- The experimental section is logical and thorough, demonstrating performance improvements across different datasets and analyzing the impact of training samples, finetuning sample size, and various design modules.

**Weaknesses:**

- Some presentation issues need improvement
    - Figure 6 should be updated to remove inappropriate "bbox" spelling marks. Additionally, all images in the paper should be replaced with vector versions to prevent blurry text, as seen in Figure 3.
- The article lacks references and discussions on self-supervised methods. The recent two works, S2Hand and HaMuCo, although not pre-training methods, also attempt to use unlabeled images and 2D off-the-shelf detectors to train 3D hand pose estimation models.
- Otherwise, the paper is relatively complete with no major weaknesses

[1] Model-based 3D Hand Reconstruction via Self-Supervised Learning

[2] HaMuCo: Hand Pose Estimation via Multiview Collaborative Self-Supervised Learning

**Questions:**

- Why are the baseline metrics relatively poor? For example, Freihand dataset shows 18+ MPJPE, while recent works (i.e. MobRecon) often achieve <6 PA-MPJPE. Could you explain if Procrustes analysis accounts for such a large performance difference? If the author could explicitly address this performance gap or more clearly explain the difference between the baseline metrics and those of existing fully supervised methods, it would be better.
- Are the positive sample augmentations identical to those used for query images?
- Is Figure 4 showing results from the FreiHand dataset?
- Regarding minibatch construction, the authors mention using 2N samples (N query images and their corresponding positive samples). Using the top-1 method for defining positive samples, could there be cases where a negative sample $I_n$ for query image $I_m$ is actually very similar but not top-1 (e.g., top-K where K>1)? Do the authors have more detailed descriptions of how to increase the discrimination in positive/negative sample sampling, or is it solely addressed through adaptive weighting?
- How is diversity ensured in the reverse lookup of top-1 samples for each query image? Could there be cases where samples from videos j and k are mutually top-1 similar samples, potentially reducing training diversity by constantly pairing samples from the same two videos?
- What specific models were used as baselines in Tab.1?

- Since the baselines compared by the author all have open-source code, in order to enhance the reproducibility of the article and the usability for downstream tasks, it is hoped that the author will adhere to what is mentioned in the article and actually release the code

---

> ### Author Response · Authors · 2024-11-23
> **Response to Reviewer 4Kod  (R2).**
>
> We appreciate the reviewer 4kod (R2) insightful and positive feedback. We have summarized the questions raised in the reviewer 4kod, as well as the areas of weakness suggested for improvement, as follows (W: Weakness; Q: Question; A: Answer):
>
> >**W1**: Some presentation issues need improvement.
>
> **A1**: Thank you very much for reviewer 4kod suggestions regarding the presentation of our work. **We have addressed the issues in the newly uploaded version.** Specifically, all images in the paper have been replaced with vector versions to ensure clarity and eliminate blurry text.
>
> >**W2**: The article lacks references and discussions on self-supervised methods. The recent two works, S2Hand[2]  and HaMuCo[3] , although not pre-training methods, also attempt to use unlabeled images and 2D off-the-shelf detectors to train 3D hand pose estimation models.
>
> **A2**: Thank you for suggesting additional related work for self-supervised methods. Although they are not pre-training methods, we find some relevance to our work. S2Hand attempts to learn 3D pose only from noisy 2D keypoints on a single-view image, while HaMuCo extends such self-supervised learning to multi-view setups. **Based on the reviewer’s recommendation, we have updated the first section of "Related Work" for discussions**. Where we have highlighted the additions in blue text between lines L110 and L114.
>
> >[2] Yujin Chen et al. "Model-based 3D Hand Reconstruction via Self-Supervised Learning."
>
> >[3] Xiaozheng Zheng et al. "HaMuCo: Hand Pose Estimation via Multiview Collaborative Self-Supervised Learning."

---

> > ### Author Response · Authors · 2024-11-23
> > **Response to Reviewer 4Kod (R2).**
> >
> > >**Q1**: Why are the baseline metrics relatively poor? For example, Freihand dataset shows 18+ MPJPE, while recent works (i.e. MobRecon) often achieve <6 PA-MPJPE. Could you explain if Procrustes analysis accounts for such a large performance difference? If the author could explicitly address this performance gap or more clearly explain the difference between the baseline metrics and those of existing fully supervised methods, it would be better.
> >
> > **A3**: To clarify first, in the Tab.1 and Tab.2, the baseline model is our fine-tuned result trained from scratch, i.e. w/o pre-training. **Then, there could be a few potential reasons for the score gap to recent works (i.e. MobRecon): 1) metric differences and 2) modeling.**
> >
> > In terms of evaluation metric, we use MPJPE, where wrist positions are aligned, to be compatible with the original studies of other fine-tuning datasets. In contrast, other works like MobRecon use PA-MPJPE, which aligns global rotation and translation between the prediction and ground-truth. This PA metric helps evaluate the local pose, but disregards the rotation error over the MPJPE. Thus, PA-MPJPE is often smaller than MPJPE.
> >
> > In modeling, there is a spectrum of backbone networks and regression schemes. Our work focuses more on validating the effectiveness of pre-training methods. Thus, similarly to PeCLR, we use the simplest modeling of 3D hand pose estimation, i.e, ResNet50 + heatmap regression that outputs 3D keypoint coordinates. In contrast, the MobRecon is based on a DenseStack backbone and a map-based position regression, which regresses both the heatmap and the position. It further utilizes the MANO model to regularize pose and construct mesh. These modeling differences account for an additional gap in performance.
> >
> > Respecting various underlying styles in modeling, we pre-train a common ResNet-50 encoder, which potentially benefits more than tailoring it to a specific architecture and makes follow-up studies more reproducible. We hope this clarification addresses the reviewer's concerns and provides a clearer explanation of the performance differences.
> >
> > >**Q2**: Are the positive sample augmentations identical to those used for query images?
> >
> > **A4**: Yes, the positive sample augmentations are identical to those used for query images, as both undergo random augmentations through various combinations in our pre-training process. Thank you for your question.
> >
> > >**Q3**: Is Figure 4 showing results from the FreiHand dataset?
> >
> > **A5**: Yes, figure 4 presents results from the FreiHand dataset. In the newly uploaded version, we have added a clarification regarding the dataset employed for the results presented in Figure 4. Thank you for your question and observation.
> >
> > >**Q4**: Regarding mini-batch construction, the authors mention using 2N samples (N query images and their corresponding positive samples). Using the top-1 method for defining positive samples, could there be cases where a negative sample In for query image Im is actually very similar but not top-1 (e.g., top-K where K>1)? Do the authors have more detailed descriptions of how to increase the discrimination in positive/negative sample sampling, or is it solely addressed through adaptive weighting?
> >
> > **A6**: We do not adopt specialized sampling techniques for positive/negative samples. Our sampling of N query images is at random from the pre-training set. This could result in cases where a negative sample is similar to a query image but not ranked as the top-1. **Yet, our adaptive weighting further helps increase discrimination between positive and negative samples, which adjusts the importance of pairs based on their similarity scores.** In the above case, our weighted value on the positive pair (top-1) is higher than the negative sample (top-K where K>1). This allows us to prioritize the feature learning for the top-1 pairs more than the rest pairs (top-K where K>1), which avoids the confusion regardless of the sample statistics of the mini-batch.

---

> > > ### Author Response · Authors · 2024-11-23
> > > **Response to Reviewer 4Kod (R2).**
> > >
> > > >**Q5**:  How is diversity ensured in the reverse lookup of top-1 samples for each query image? Could there be cases where samples from videos j and k are mutually top-1 similar samples, potentially reducing training diversity by constantly pairing samples from the same two videos?
> > >
> > > **A7**: We appreciate providing us with an insightful test case. When “samples from videos j and k are mutually top-1 similar samples” as the reviewer suggested, we can derive a trivial case where the two videos (j and k) capture almost the same activities with the identical hand poses and camera angles. Then we can ensure it is very unlikely if these collected videos are curated from different sources on the Web (like 100DoH) or made by asking unique participants to behave without any scripted instructions (like Ego4D). **As such, enriching the diversity of subjects, performing tasks, and captured environments serves to avoid such unintended consequences in sample pairing. This is also the goal of our work, which aims to pre-train on large-scale, real-world hand data.**
> > >
> > > >**Q6**: What specific models were used as baselines in Tab.1?
> > >
> > > **A8**: The baseline model used in Table 1 is based on ResNet50 + heatmap regression, which is referred to as minimal-hands [4]. **A more detailed explanation of the architecture for 3D hand pose estimation can be found in Section 6.4  of the supplementary materials.**
> > >
> > > >**Q7**: Since the baselines compared by the author all have open-source code, in order to enhance the reproducibility of the article and the usability for downstream tasks, it is hoped that the author will adhere to what is mentioned in the article and actually release the code.
> > >
> > > **A9**: We once again express our gratitude for reviewer 4kod appreciation of this work. We understand the importance of releasing the code to enhance the reproducibility of the paper and to promote its applicability in downstream tasks. **We plan to release our code, checkpoints, and also pre-processed assets (e.g., hand bboxes and frame indices from Ego4D and 100DOH, 2D keypoints and similarity scores) corresponding to 2 million large-scale wild hand images upon publication.**
> > >
> > > **Reference**:
> > > >[4] Yuxiao Zhouet al. "Monocular real-time hand shape and motion capture using multi-modal data."

---

> > > ### Comment · Reviewer_4Kod · 2024-11-23
> > >
> > > Thank the author's detailed and positive response.
> > >
> > > I now have a clearer understanding of the performance differences caused by MPJPE, PA-MPJPE, and the model structure. However, to more rigorously evaluate the performance of the HandCLR pre-trained model on downstream tasks—not with the goal of achieving SOTA, but to establish a reliable baseline for comparison—I would like to suggest that the authors report the baseline performance of the model using PA-MPJPE as the evaluation metric.
> > >
> > > At least, including this in the experiments presented in Table 1 would provide a more comprehensive understanding of the improvements achieved by the proposed model over current methods (e.g. MobRecon) that do not utilize large-scale pretraining.

---

> > > > ### Author Response · Authors · 2024-12-03
> > > > **Response to Reviewer 4Kod (R2).**
> > > >
> > > > Dear Reviewer 4Kod (R2),
> > > >
> > > > We thank the reviewer 4Kod (R2) for the understanding of the metric and model designs and helpful suggestions. We promise that we will include the scores of PA-MPJPE on FreiHAND to make it easier to compare with public baselines in the final version.
> > > >
> > > > To further consolidate our contribution regarding method comparisons, we have addressed additional comparisons suggested by the rest of the reviewers. These include comparisons with other public estimation methods, such as A2J [5], Spurr et al. [6], SVEgoNet [9] etc. (**R3-Q2**), a video contrastive learning method, TempCLR (**R1-W1**), and a weakly supervised setting (**R3-Q1**).
> > > >
> > > > Once again, we appreciate the reviewer 4Kod (R2)'s time and valuable comments.
> > > >
> > > >
> > > > Best regards,
> > > >
> > > > All Anonymous Authors

---

### Official Review · Reviewer_vbov · 2024-11-06

**Soundness:** 3
**Presentation:** 3
**Contribution:** 2
**Rating:** 6
**Confidence:** 4

**Summary:**

This paper presents a contrastive learning method for the pre-training of 3D hand pose estimation based on large-scale in-the-wild hand images. A parameter-free adaptive weighting mechanism is introduced in the contrastive learning loss, which not only learns from similar samples but also adaptively weights the contrastive learning loss based on inter-sample distance. Experiments show improved performance compared with existing pre-training methods.

**Strengths:**

- The paper is well written and easy to follow.
- The motivation of finding similar hands derived from different video domains is technically sound, which can further benefit contrastive learning process from discriminating foreground hands in varying backgrounds.
- The experimental results in Table 3 demonstrate the generality of the proposed contrastive learning with adaptive weighting mechanism.

**Weaknesses:**

- TempCLR [1] proposes a pre-train framework for 3D hand reconstruction with time-coherent contrastive learning, and shows better performance compared with PeCLR. Although TempCLR focuses on reconstruction tasks, the used parametric model can output 3D pose results. Therefore, more comparisons with TempCLR would be helpful.

- In the second column of Figure 6, HandCLR demonstrates advanced performance in hand-object occlusion. Does the proposed method exhibit robustness in similar severe occlusion scenarios involving hand-object interactions? More qualitative analysis in datasets like DexYCB or in-the-wild scenarios would be helpful.

- The proposed adaptive weighting mechanism is a straightforward approach that has proven effective; however, it lacks a clear articulation of its motivation, particularly regarding the challenges faced in 3D hand pose estimation tasks as mentioned in the introduction.

[1] Tempclr: Reconstructing hands via time-coherent contrastive learning. In  3DV, 2022.

**Questions:**

N/A

---

> ### Author Response · Authors · 2024-11-23
> **Response to Reviewer vbov (R1).**
>
> We sincerely appreciate the careful and thoughtful comments and the time reviewer vbov (R1) spent on them, **especially the suggestion to compare with the TempCLR method**. We have provided responses to all the questions as follows (W: Weakness; Q: Question; A: Answer):
>
> > **Q1 (W1)**: TempCLR [1] proposes a pre-train framework for 3D hand reconstruction with time-coherent contrastive learning, and shows better performance compared with PeCLR. Although TempCLR focuses on reconstruction tasks, the used parametric model can output 3D pose results. Therefore, more comparisons with TempCLR would be helpful.
>
> **A1**:  Thank you for your suggestion. **We have added a comparison with the experimental results of TempCLR [1]**. As shown in the Table below, we evaluate our comparison methods pre-trained from the 50K and 100K sets of Ego4D and fine-tuned on FreiHands*. The TempCLR’s performance surpasses PeCLR, which is consistent with the results reported in the original paper.
>
> | Method | Pre-training size | MPJPE (↓) | PCK-AUC (↑) |
> |----------|----------|----------|----------|
> | PeCLR | Ego4D-50K | 47.42  | 49.85 |
> | TempCLR | Ego4D-50K | 45.17 | 52.40 |
> | **HandCLR** | **Ego4D-50K** | **35.32** | **63.35** |
> |   |   |   |   |
> | PeCLR | Ego4D-100K | 46.00 | 51.50 |
> | TempCLR | Ego4D-100K | 44.54 | 53.28 |
> | **HandCLR** | **Ego4D-100K** | **31.06** | **68.66** |
>
> We find the following limitations of TempCLR compared to our method:
> * 1) inefficiency in data collection
> * 2) limited gains in contrastive learning from neighbor hand images.
>
> **While TempCLR treats neighbor hand frames as positives, the tracklets of hands in such dynamic egocentric videos are often truncated due to hand occlusion or hand detection failures.** This makes it difficult to collect hand crops in adjacent frames. Indeed, its collection requires _x4_ more images of detected hand crops to construct 100K samples of neighbor hands from Ego4D, which suggests that detected hand images are not fully utilized.
>
> Furthermore, as shown in Figure 2, the sampled neighbor frames often have limited diversity in backgrounds. Thus, the improvement over PeCLR, which makes positive pairs from a single image, is marginal. In contrast, our HandCLR leverages similar hands that provide diverse characteristics, including various types of
>
> * 1) hand-object interactions,
> * 2) backgrounds, and
> * 3) appearances.
>
> That's the reason HandCLR exhibits significant gains over PeCLR and TempCLR.
>
> > **Q2 (W2)**: In the second column of Figure 6, HandCLR demonstrates advanced performance in hand-object occlusion. Does the proposed method exhibit robustness in similar severe occlusion scenarios involving hand-object interactions? More qualitative analysis in datasets like DexYCB or in-the-wild scenarios would be helpful.
>
> **A2**: Yes, our method indeed demonstrates enhanced robustness in hand-object occlusion scenarios.
> Our HandCLR method benefits from pre-training on large-scale in-the-wild videos, including complex and various hand-object interactions. **In addition, our pre-training from non-identical similar hands effectively handles scenarios where the query image contains partial occlusion, while the corresponding similar hand image does not, and vice versa. Such examples can be found in both Figure 2 and Figure 7.**
> We appreciate the reviewer’s recommendation to include more qualitative analyses. In the appendix of the revised paper, we will add additional visualizations in hand-object interaction scenarios, such as DexYCB.
>
> > **Q3 (W3)**: The proposed adaptive weighting mechanism is a straightforward approach that has proven effective; however, it lacks a clear articulation of its motivation, particularly regarding the challenges faced in 3D hand pose estimation tasks as mentioned in the introduction.
>
> **A3**: The motivation behind our adaptive weighting mechanism lies in effectively utilizing similar hands in the contrastive learning framework. A naive approach to using sampled similar hands in the contrastive learning is to simply replace original positive pairs with the similar hands, but it fails to account for the degree of similarity between pairs. Our adaptive weighting scheme overcomes this limitation by dynamically assigning higher weights to more similar pairs, enabling the model to better capture the proximity of samples and enhance the contrastive learning.
>
> **In response to reviewer vbov’s comments, we have updated the introduction in the newly uploaded version**. Please refer to the L81-L87 in the newly uploaded version, where the additions are highlighted in blue.
>
> **Reference**:
> >[1] Andrea Ziani et al. "Tempclr: Reconstructing hands via time-coherent contrastive learning."

---

> > ### Comment · Reviewer_vbov · 2024-11-26
> >
> > The authors provide a comparison with TempCLR, which addresses previous concerns. Meanwhile, considering the comments of other reviewers and the author's responses, my final rating is: 6: marginally above the acceptance threshold

---

> ### Author Response · Authors · 2024-11-28
> **Response to Reviewer vbov (R1).**
>
> Dear Reviewer vbov (R1),
>
> We would like to sincerely thank you for reviewer vbov's thoughtful feedback and for acknowledging the improvements in our work, particularly in addressing the concerns raised by previous reviewers. Your recognition of our efforts means a great deal to us.
>
> We would like to kindly remind you that ICLR allows for the modification of ratings directly within the reviewer's system. This ensures that the rating changes are effectively recorded and considered. We would greatly appreciate your assistance in updating the rating directly in the reviewer's system to reflect the improvement of your rating.
>
> Once again, thank you for reviewer vbov's time and valuable comments.
>
>
> Best Regards,
>
> All Anonymous Authors

---

### Author Response · Authors · 2024-11-30
**Global response (Summary of the rebuttal period)**

**We would like to thank the area chairs and reviewers for their efforts and suggestions in reviewing our paper**.  We have revised the manuscript according to the reviewers' suggestions and **marked all important modifications in blue** in the new version of the submission. **The important differences include** (W: Weakness; Q: Question):
* **Motivation Clarifications**: Following Reviewer vbov (R1)'s Weakness 3 (W3), we have updated the section on "adaptive weighting" in the introduction section to address the comment regarding the "lack of a clear expression of motivation."
* **Detailed Related Work**: Following Reviewer 4Kod (R2)'s Weakness 2 (W2), we have updated the related work section by incorporating discussions of two relevant studies: S2Hand and HaMuCo, thus enhancing the discussion on self-supervised methods.
* **More Visualization**: Following Reviewer vbov (R1)'s Weakness 2 (W2), we have revised the visualization part in the experiment section by adding a discussion on Hand-Object occlusion and providing visual results on the DexYCB dataset.

As minor revision, the experimental text descriptions have been accordingly updated based on the reviewers' suggestions and subtly adjusted to align with these modifications:
* **R2-Q6, R3-Q7, R3-Q8**: Changed “baseline” to “w/o pre-training” in Tab.1 and Tab.2 for clarity, and improved result notation by using bold for the best and underlining for the second-best results. Captions were updated accordingly.
* **R2-Q3**: Revised Fig.4's caption to include the FreiHand dataset.

Additionally, **we have provided the requested experimental results** in response to the reviewers' concerns about Weaknesses & Questions **(R1-W1, R3-Q1,2,3)**. **During the rebuttal period, the additional experiments we included were as follows**:
* **TempCLR Comparison**: To address Reviewer vbov's (R1) Weakness 1 (W1), we provided a comparison of TempCLR under different pre-training scales. Additionally, we explained the challenges we encountered when applying TempCLR to large-scale wild hand data.
* **WSL Setting Comparison**: To address Reviewer PnQd's (R3) Question 1 (Q1), we provided comparison results in the weakly supervised setting. We also explained the differences between the weakly supervised learning (WSL) setup and pre-training.
* **3D HPE Method Comparison**: To address Reviewer PnQd's (R3) Question 2 (Q2), we provided a comparison of the dataset with other 3D hand pose estimation (HPE) methods.
* **MSE Loss Comparison**: In response to Reviewer PnQd's (R3) Question 3 (Q3), we provided the results of a model pre-trained using MSE loss and subsequently fine-tuned it.

**We will release the data, code, and pre-processed assets (e.g., hand bounding boxes, frame indices from Ego4D and 100DOH, 2D keypoints, and similar hand labels) corresponding to 2 million (2.0M) large-scale wild hand images to promote the development of the research community.**

In conclusion, we sincerely thank the area chairs and reviewers again for their valuable feedback, which has significantly improved the quality of our work.

---

### Meta-Review · Area_Chair_kJr4 · 2024-12-20

**Metareview:**

This paper proposes a contrastive-learning method for pre-training of 3D hand pose estimation based on large-scale in-the-wild data.

The three reviewers all appreciate the straightforward nature and effectiveness of the approach, especially as it is demonstrated on large-scale data.

Some weaknesses raised include issues of presentation, as well as discussion and placement of the proposed method with respect to the larger body of literature on self-supervised learning.  This has largely been addressed through the author response.

The AC has read through the reviewer comments and author responses.  Given the strong support by all the reviewers (6,6,8), the AC recommends that the paper be accepted.  The authors are requested to incorporate the content in their response to reviewers to their camera-ready version of the paper.

**Additional Comments On Reviewer Discussion:**

During the discussion period, the authors provided additional discussion and comparisons to existing works on self-supervised learning as well as experimental comparisons to other related works.  The reviewers appreciate the added efforts.

---

### Decision · Program_Chairs · 2025-01-22

Accept (Poster)